# Some Important Issues of the Commercial Production of 1-D Nano-PANI

**DOI:** 10.3390/polym11040681

**Published:** 2019-04-15

**Authors:** Ying Wu, Jixiao Wang, Bin Ou, Song Zhao, Zhi Wang

**Affiliations:** 1CERC, School of Chemical Engineering and Technology Tianjin University, Tianjin 300354, China; wuying19901213@163.com (Y.W.); binou@tju.edu.cn (B.O.); zhaosong@tju.edu.cn (S.Z.); wangzhi@tju.edu.cn (Z.W.); 2Tianjin Key Laboratory of Membrane Science and Desalination Technology, Tianjin University, Tianjin 300354, China; 3State Key Laboratory of Chemical Engineering, Tianjin University, Tianjin 300354, China; 4Collaborative Innovation Center of Chemical Science and Engineering (Tianjin), Tianjin University, Tianjin 300354, China

**Keywords:** one-dimensional polyaniline nano-materials, commercial production, synthesis methods, waste treatment

## Abstract

One-dimensional polyaniline nano-materials (1-D nano-PANI) have great promise applications in supercapacitors, sensors and actuators, electrochromic devices, anticorrosive coatings, and other nanometer devices. Consequently, commercial production of 1-D nano-PANI at large-scale needs to be quickly developed to ensure widespread usage of this material. Until now, approaches—including hard template methods, soft template methods, interfacial polymerization, rapid mixing polymerization, dilute polymerization, and electrochemical polymerization—have been reported to be used to preparation of this material. Herein, some important issues dealing with commercial production of 1-D nano-PANI are proposed based on the complexity of the synthetic process, its characters, and the aspects of waste production and treatment in particular. In addition, potential solutions to these important issues are also proposed.

## 1. Introduction

Polyaniline (PANI) has been known as ‘aniline black’ for more than a century [1]. Today, PANI is one of the most investigated and most promising conducting polymer for potential commercial applications. Over the past four decades, much attention has been paid on intensive research on PANI including its preparation, characterization, and applications in sensors and actuators, electromagnetic shielding, conductive paints, energy storage components, protective coatings, photoelectric devices, and separating membranes. These applications are attributed to its simple preparation process, a cost-effective monomer, tunable properties, good environmental stability, unique acid and redox doping, electrochemical properties, reversible redox behavior, high machinability, and high conductivity [2,3,4,5,6,7,8]. At present, bulk polyanilines are commercially produced by chemical oxidation process, and are mainly used in anticorrosive coatings, antistatic coatings, etc. The companies producing them are all over the globe, including Ormecon (Germany), Ancatt (USA), Zhongke Benan (Hunan, China), Dupont and its subordinates Vniax company, Allied company, IBM company, Zipperling Kessler (Germany), Neste Oy (Finland), and NittoDenko (Japan). Many other companies are actively developing the industrial application of polyaniline.

The ease of dispersion of PANI in solution [9], its short diffusion path, large specific surface area [10], and high one-dimensional electrical conductivity provides that 1-D nano-PANI including nanofibers, nanowires, nanobelts, nanotubes, nanorods, and nanoneedles have superior properties to its bulk counterpart and exhibits great promise for use in sensors and actuators [11,12,13], electrochromic devices [14,15], supercapacitors [16,17,18], protective coatings [19,20], and other nanometer devices [21,22]. However, large-scale commercial production of 1-D nano-PANI is a critical issue in determining its wide scale applications.

Although many approaches, including those listed above, have been established, the preparation of PANI—for example 1-D nano-PANI with aimed morphology and properties—is not quite as simple as one think. Its complexity results from its linkage or growth patterns and other parameters. First of all, although attacking (linking) the *para*-position of the aniline monomer has usually been represented as the process for formation of PANI. In fact, during the formation of PANI, it is also will attack (link) at *ortho*- and *meta*-position of the aniline monomer. This process will make the PANI macromolecular chain branch out, which will deteriorate the conjugation of the molecular chain. The degree of branching out may be affected by polymerization temperature, doping acid, and other parameters. Secondly, the degradation PANI and aniline oligomers always accompanied its polymerization and the degraded products can participate in the PANI polymerization process. In order to obtain 1-D nano-PANI, the products must be purified. During this process, a large amount of waste liquid—which may contain aniline, oligomers, and degradation products—is certainly discharged. Normally, the discharged waste liquid is heavily black, high concentration acid and bactericide. Therefore, the treatment of this waste liquid is tedious, difficult, and expensive. Usually, a powerful oxidation method like the Fenton process is needed for complete treatment [23,24,25]. From this, we can conclude that the most crucial aspect of the commercial production of 1-D nano-PANI is the requirement of the reduction of waste treatment cost and simplification of the waste liquid treatment process. Besides waste treatment, there are other issues in the process that also limit commercial production, such as the cost of the method and problems in scaling up. This review discusses the complexity of the PANI synthetic process, advantages, and disadvantages of the various synthesis methods for production of 1-D nano-PANI.

## 2. Complexity of the Polymerization Process of PANI

The production process of PANI is quite complex. Two kinds of side reactions occur during the polymerization of aniline. The degree of side reactions is determined by reaction conditions, which ultimately affects the morphology and property of PANI and the composition of the discharged waste water.

The first kind is that aniline is not only linked at the *para*- position of aniline (head–tail connection type), but also linked at *ortho*- and *meta*- positions. The cross-linking and branching out of the PANI macromolecular chain (Scheme 1) is determined by the polymerization conditions such as reaction temperature. When the linking happens at the *ortho*- and *meta*- position, the conjugation degree of PANI is reduced by steric effects, and hydrophilic amido- and imino- groups are hidden while the hydrophobic phenyl groups along the PANI polymer are protruded. These effects cause a decrease in their electronic conductivity and hydrophilicity [26]. When polymerization carried out under conditions of pH < 2, lower temperature, and lower oxidation potential, it is beneficial for *para*- position linkage among aniline monomers, which is due to the higher activation energy of the side reactions [27,28,29]. Meanwhile, PANI at a high oxidation state has the ability to oxidize aniline monomers to form oligomers, which make the changes of PANI properties.

The second kind of side reaction is the hydrolysis of imines and the Michael addition between side products benzoquinone (BQ) and amine group [30]. Through the polymerization process, hydrolysis reaction always takes place and BQ generate by hydrolysis of imines parts of PANI. The formed BQ will react with the amine group of aniline, aniline oligomers, and PANI through Michael addition reaction [30,31,32]. The by-products—including co-oligomers of aniline and *p*-benzoquinone (CAB), BQ, and hydroquinone (HQ)—generated during this reaction will accumulate through side reactions (Scheme 2) [32]. These by-products will influence the purity, morphology, and properties of the resultant PANI. This process, resulting in composition of the discharged waste liquid, is very complex. The composition of the waste liquid which was generated by electrochemical polymerization of 1-D nano-PANI was characterized by high performance liquid chromatography coupled to photodiode diode array detection (HPLC–DAD) and ultraviolet and visible (UV–vis) spectrophotometer, and the chromatograms were shown in the Figure 1 [32]. In the HPLC chromatograms, the peaks 1 to 7 belong to HQ, BQ, aniline, *N*-phenyl-1,4-benzoquinonediimine (PBQD) [33,34], and *N*-phenyl-1,4-benzoquinonemonoimine (PBQM) [35], *p*-aminodiphenylamine (PPD) [36], 2,5-dianilino-*p*-benzoquinone (DABQ) [37], and Peaks 8 and 9 belong to the CAB with larger molecular weight which has similar structure with DABQ. PPD and PBQD represent reduction state and oxidation state of the aniline dimmers, respectively. As the oxidation potential and reaction temperature elevated, the imines concentration in the solution increases. As a result, the hydrolysis reaction rate increases, this leads to an increase of the concentration of BQ. As a result, the amount of by-products HQ, BQ, and CAB increases in the reaction solution, which affects the properties of PANI through copolymerization (Scheme 3).

When aniline is polymerized via chemical process, the compositions of discharged waste liquid are much more complex than that of waste liquid obtained via the electrochemical process. That is to say that, besides components such as mentioned above, others such as the oxidants and the templates used in the synthesis of 1-D nano-PANI will co-exist in the discharged liquid waste. The difficulty in treatment and recycling of the waste liquid is a major impediment to the commercial production of 1-D nano-PANI. In addition, the side reactions in this process have an adverse effect on the properties of the 1-D nano-PANI.

## 3. Characters of the Various Synthesis Methods of 1D Nano-PANI

In this part of review, some characters related to the most available approaches for preparing 1-D nano-PANI will be discussed. According to the method to initiate polymerization, the synthesis method can be divided into two kinds of processes: chemical polymerization and electrochemical polymerization [38]. According to whether templates are used or not, the synthesis methods can be divided into three kinds of processes, which are hard template methods [39], soft template methods [40], and no template methods. No template methods include interfacial polymerization [41,42], rapid mixing polymerization [43], dilute polymerization [44], seeding polymerization [45], high gravity chemical oxidative polymerization [46], etc. Other methods used for 1-D nano-PANI production—including sonochemical synthesis [47,48], solid-state polymerization [49,50], UV light-assisted polymerization [51], radiolytic synthesis [52,53], microwave-assisted polymerization [54], plasma-induced polymerization [55,56], electrospinning [57], etc.—are also discussed.

### 3.1. Hard Template Methods

The hard template synthesis method for 1-D nano-PANI was first reported by Martin et al. [58,59,60,61]. There are two main types of hard templates, including porous materials and 1-D nanomaterials. Porous materials include track etched membranes [60,61], zeolites [62], anodic aluminum oxide (AAO) [63,64,65,66], TiO_2_ nanotubes array [66], meso-porous silica [67], etc. 1-D nanomaterials include silica nanotubes [68], halloysite nanotubes [69,70], manganese oxide 1-D nanomaterial [71], copper wires or rings [72], electrospun fibers [73], carbon nanotubes(CNTs) [74], amyloid nanofibers [75] and thin-glass tubes [76], nanochannels [77], etc. The synthesis process of 1-D nano-PANI through hard template methods is presented in Scheme 4.

With respect to the various porous materials, such as track etched membranes and AAO, the diameter and depth of the pore determine the eventual morphology of the synthesized 1-D nano-PANI, and normally the obtained PANI protruding from the template are not one-dimensional nano-shaped. In the case of oriented and uniform pore structure, two typical porous materials—AAO and track etched membranes—have received much more attention as templates for preparing PANI. Tracked-etched polymer membranes contain cylindrical pores that are randomly distributed across the membrane surface with a uniform diameter and a wide choice of pore diameters (to as little as 10 nm), and pore densities approaching 10^9^ pores/cm^2^ [60]. However, the increase of pore densities will induce partial overlap of the pores, which will produce uneven polyaniline nanotube morphology. AAO porous membranes have an oriented porous structure with uniform and nearly parallel pores with a wide average pore diameter of 5–100 nm (up to 200–300 nm after additional pore widening process) and pore density of 10^11^–10^8^ pores/cm^2^ [63,78,79].

In the case of the 1-D nanomaterial templates, most of these templates are used to produce 1-D nano-PANI composites [74,75,76,79], but some templates like manganese oxide 1-D nanomaterials can be sacrificed themselves during the polymerization of aniline [71].

The character of the hard template synthesis methods is that, by using templates with oriented and uniform structures—such as track etched membranes [61], AAO [63,79], and oriented manganese oxide 1-D nanostructures—directionally aligned 1-D nano PANI material can be produced which can be used in field emission devices and supercapacitors [39]. Sometimes, under controlled conditions, PANI with a nanotube shape will be formed, which might be beneficial for improving the performance of electrical devices, such as supercapacitors and sensors.

The drawback of the hard template method might be the high cost of the templates, tedious pre-or post-treatment, and complex waste liquid treatments. Usually, the hard templates are expensive and only can be used once. For example, track etched membranes are prepared by accelerated heavy ion beam irradiation and a chemical etching process. AAO templates are prepared by an anodic oxidation treatment of high purity aluminum in acid. To obtain through holes in the AAO film, it must be separated from the aluminum substrate. Meanwhile, mesoporous silica and manganese oxide 1-D nanomaterials are usually prepared by hydrothermal methods. The synthetic methods used to prepare the hard template are responsible for their high cost and a large amount of waste liquid discharge. For the porous material, considering that the nanofiber can only be grown in pore, the pore is only a small part relative to the whole volume of the porous material and the reaction solution, so the yield of the one-dimensional nanometer polyaniline material is low. If pure 1-D nano-PANI wants to be obtained, the hard templates such as track etched membranes and AAO templates must be dissolved in a certain solution. As a result, recycling of the hard template is extremely difficult. The amount of waste that needs to be disposed of is very large, including aniline (low yield) and waste resulting from hard template synthesis and polyaniline release [80]. Another drawback might be that, during the template dissolving process, the 1-D nano-PANI may be destroyed or may form undesirable aggregated structures.

In a few words, due to the cost of the hard templates, tedious pre-/post-treatment and waste liquid treatment, the hard template synthesis process is too expensive and complex to be a good selection method to be used for commercial production of 1-D nano-PANI at large scale.

### 3.2. Soft Template Methods

Soft template methods are also known as self-assembly methods. Many structural direct agents, which can act as soft templates based on their self-assembly, have been reported for preparation of 1-D nano-PANI. The soft templates are usually amphiphatic molecules, including surfactants, block copolymers, biomolecule, etc. Surfactants include sodium dodecylbenzenesulfonate (SDS) [81,82], cetyltrimethylammonium bromide (CTAB) [83], 1-butyl-3-methylimidazolium chloride([bmim]Cl) [84], CTAB gels [85], CA-PEG400 gels [86], dodecylbenzenesulfonic acid(DBSA) [87,88] naphthalene sulfonic acids (NSA) [80,89,90,91], *p*-toluenesulfonic acid (p-TSA) [92], aminobenzenesulfonic acid (SAN) [93], amino acids [94,95] 3,5-dinitrosalicylic acid [96], camphorsulfonic acid (CSA) [97], pyrene sulfonic acid(PSA) [98], saturation fatty acids [99], hydrogensulfated fullerenol of six -(O)SO_3_H groups (C_60_(OSO_3_H)_6_) [100], sulfonated dendrimer with 24 terminal groups of 3,6-disulfonaphthylthiourea (PAMAM4.0[naphthyl(SO_3_H)_2_]_24_) [100], etc. Block copolymers include Triton X-100 [101], poly(styrene-block-4-vinylpyridine) (PS-*b*-P4VP) [102], etc. Biomolecule include sucrose ester [103], sodium alginate [104,105], cyclodextrin [106,107,108], heparin [109], vitamin C [110], chitosan [111], DNA [112,113,114], citric acid [11], tobacco mosaic virus RNA [115,116,117], etc. Other amphipathic molecules also can be used as soft templates, such as crown ethers [118], dye [119], and urea [13].

With the presence of surfactants in the aniline polymerization system, the mechanisms of soft template methods involve the self-assembly of surfactants under the condition of aniline polymerization. Under certain condition, surfactants will assemble into tubular micelles, these materials include NSA [80], Triton X-100 [101], tartaric acid [120], etc. Gels, such as (CTAB and lauric acid) [85] and CA-PEG400 [86], have a three-dimensional network structure. Other templates—such as cyclodextrin [106], methyl orange (MO) [121], and vitamin C [110] —appear to self-assemble to 1-D nanostructures. The synthesis process of 1-D nano-PANI through soft template methods can be seen in Scheme 4.

The materials used to assemble soft templates might be separated from the 1-D nano-PANI by washing processes, so no other tedious post process is needed. Furthermore, most of the soft template materials are less expensive than those of the hard templates, and there are a wide range of soft template materials that can be chosen. The cost of soft template method is lower than that of the hard template method. Furthermore, the morphology and properties of the 1-D nano-PANI can be controlled by varying the soft template materials and the synthesis conditions. For example, Wanet et al. controlled the average diameter, crystallinity, and conductivity of the synthesized PANI nanofibers and nanotubes by changing either the chain lengths of dicarboxylic acids (HOOC(CH_2_)*_n_*COOH) or the molar ratio of the acids to the aniline monomer [122]. Others have also found that the average diameter, pore shape, and conductivity of the PANI can be controlled, through changing the type of cyclodextrins and the molar ratio of the acid to the aniline monomer [106,107,123]. Jayakannan et al. found that varying the molar ratio of the acid to the aniline monomer and the solvent can produce various 1-D nano-PANI morphologies from nanofibers to nanotapes [73]. Furthermore, the yield of 1-D nano-PANI is relatively high in these methods [124].

However, there are drawbacks in producing 1-D nano-PANI by soft template methods at large commercial level. The first is that the discharged waste water from soft template is complex, some materials used to induce formation of 1-D nano-PANI exist in the discharged waste waters. Normally, discharged waste waters from the most common soft template methods for 1-D nano-PANI synthesis contain soft template materials (Scheme 5), aniline, BQ, HQ, aniline oligomers, CAB, acids, and salts. Under conventional conditions, it is difficult to recycle the soft template materials because of their high solubility. This situation will greatly increase the cost of treating waste waters. The second drawback is that some soft template materials, such as lipid, MO, DNA, tobacco mosaic virus, heparin, sodium alginate, PSA, PAMAM4.0[naphthyl(SO_3_H)_2_]_24_, are too expensive to be suitable for a commercial production 1-D nano-PANI at large scale.

Therefore, it can be concluded that soft template approaches might not be a feasible method to produce 1-D nano-PANI at large scale due the associated cost of waste treatment.

### 3.3. Chemical Polymerization with Template-Free Approaches

Chemical polymerization with template-free approaches includes interfacial polymerization, rapid mixing methods, dilute polymerization, and high gravity chemical oxidative polymerization. The mechanism of template-free approaches is based on preventing the “secondary growth” of PANI, which is caused by heterogeneous nucleation or the stacking of nucleates though Brownian movement [125]. PANI preferentially forms nanofiber morphology in the aqueous phase during homogeneous nucleation, and large agglomerates with irregularly shaped particles can result from secondary growth [126,127]. The characters of interfacial polymerization, rapid mixing methods, dilute polymerization, and high gravity chemical oxidative polymerization will be discussed in this part of the review.

#### 3.3.1. Interfacial Polymerization

Interfacial polymerization method used for preparing 1-D nano-PANI was first reported by Kaner et al. [128]. In the interfacial polymerization method, polymerization occurs only at the interface of the organic/aqueous interphase. The formed PANI will depart from the interface at its initial stage, where no reactants have and the polymerization should be terminated [129], which prevents any “secondary growth”. As a result, PANI nanofibers are formed almost exclusively. Until now, organic solvents—such as benzene, chloroform, toluene, hexane, xylene, diethyl ether, carbon disulfide, carbon tetrachloride, *o*-dichlorobenzene, or methylene chloride—have been used to separate solution containing aniline from a solution containing oxidants [41,128,129,130,131]. The liquid/liquid interface, solid/liquid interface using porous membrane, and gas/liquid interface employing CO_2_ or aniline vapor are currently under research for the production of pure PANI nanofibers [132,133,134]. A microemulsion interface can also be used to produce 1-D nano-PANI [135]. The synthesis process of 1-D nano-PANI through interfacial polymerization is presented in Scheme 6.

The PANI product prepared by interfacial polymerization method is in inhomogeneous state. The commonly used oxidant ammonium persulfate (APS) is a strong oxidant, its standard redox potential is 2.05 V vs. NHE. In the interfacial polymerization process, the water phase has a high redox potential at the initial stage of the reaction, which can produce a large amount of nuclei and a high degree of by-products. The large quantity of nuclei leads to collision and stacking of the polymer chains, and this induces “secondary growth”. As a result, the PANI aggregates quickly to form irregular particles. The side reactions also include reactions at the *ortho*- and *meta*- positions as well as imine hydrolysis reactions that affect the morphology and properties of PANI. An increase in the initial oxidation potential of the solution will cause the PANI products having more irregular morphology, electronic conductivity, and lower purity. Wang et al. found that with the increase of the APS to aniline molar ratio from 1/6 to 1/1, more irregularly shaped agglomerates generated and the electronic conductivity decrease from 8.6 to 0.16 S/cm [136].

Changing the aniline concentration in solution from 0.032 to 1.6 M produced no significant effect on the observed morphology of the polymer product [129]. Increasing the aniline concentration in the reaction solution reduced the volume of organic solvent, which increased the rate of the polymerization reaction. This approach offers great potential for scaling up the production of 1-D nano-PANI. The main problems associated with the interfacial polymerization method in terms of a large commercial scaling of 1-D nano-PANI production include environmental pollution by organic solvents, tedious waste treatment, and low yield. At first, volatile organic compounds in the production process lead to environmental pollution. In this regard, the waste liquids have two phases, an organic phase and an aqueous phase. Normally, the organic phase includes aniline, HQ, BQ, and oligomers which make the waste treatment processes tedious and costly. In addition, the yield of 1-D nano-PANI from the interfacial polymerization process is low, because typically the aniline to APS molar ratio is maintained at 1/4 [128,129,131]. To obtain a 100% yield, the theoretical molar ratio of APS to aniline should be at about 1.2–1.25. Weiller et al. found that the yield of PANI from their interfacial polymerization process was about 6%–10% under the condition of APS to aniline molar ratio maintained at 1/4 [128]. Since more than 90% of the aniline in the reaction solution does not participate in the reaction, so, if 1 kg PANI wants to be produced, about 9.25 kg of aniline must be degraded or recycled, which is calculated based on the emeraldine (EB) form. Although some research into increasing 1-D nano-PANI yields has been donbe, the low yield of 1-D nano-PANI product using the interfacial polymerization process remains a problem [42,136]. Together with the cost of waste treatment, the commercialization of the interfacial polymerization process for large scale production of 1-D nano-PANI is not the most feasible.

#### 3.3.2. Dilute Polymerization

Dilute polymerization, first reported by Epstein [44]—typically polymerized at aniline monomer concentrations below 10^−3^ M—can prevent collision and stacking of nucleates by Brownian motion which favors the growth of the 1-D nano-PANI [44,136,137,138,139].

The advantages of dilute polymerization are that templates are not needed and that the reaction solution is relativity simple in composition. The major disadvantage of dilute polymerization is the huge quantity of equipment and large volume of waste water. The typical conditions for dilute polymerization are concentrations of aniline and doping acid of 8 mM and 1.0 M, a reaction time of 24 h, and an APS-to-aniline molar ratio maintained at 1/2, which was first reported by Epstein et al. Under these conditions, the theoretical maximal yield of 1-D nano-PANI is about 40%, because 1 mole of aniline will consume 1.25 mole of APS to become emeraldine salt form of polyaniline. Assuming that the yield of 1-D nano-PAN is 40% and the doping acid is sulfuric acid, the volume of equipment and waste water can then be calculated. In order to obtain 1.0 kg 1-D nano-PANI by this method—as calculated in the EB form—3453.2 L reaction solution, and 24 h of reaction time would be required. At the completion of polymerization, there is about 60% aniline monomer, about 1.0 M acid, 4 mM ammonium sulfate, and trace quantities of HQ, BQ, and aniline oligomers co-exist in the reaction solution which will be discharged as waste liquid. Due to the low concentration of raw materials and reaction rate, the volume of equipment is huge, which is costly. The total waste water would include the reaction solution plus others such as those used to wash and purify the product. The wet 1-D nano-PANI, obtained in EB form from filtration, normally contains at least 90% reaction solution, which is the solution obtained following polymerization. In order to obtain dry 1-D nano-PANI product at high purity, excessive aniline monomer, acid, ammonium sulfate, and trace quantities of by-products—including HQ, BQ, and aniline oligomers—must be washed out. The density of 500 mM sulfuric acid would be 1.03, the density of solution after polymerization would be approximately 1.03. Therefore, the 1.0 kg dry 1-D nano-PANI product would contain 0.438 kg of impurities which would include the aniline, ammonium sulfate, and sulfuric acid in the solution after polymerization. With three washings, at least 172.6 L of deionized water would be needed to reduce the impurity content of the 1.0 kg of 1-D nano-PANI product to 0.001. The total volume of waste water from the process would be at least 3625.8 L, which would be expensive to treat. In some reports, the concentration of aniline was increased to 0.01 M and the APS-to-aniline molar ratio was increased to 1 [137,139]. These conditions will increase the yield of PANI and will reduce the concentration of aniline in the waste solution after polymerization. This would decrease the volume of the equipment needed as well as the waste water treatment, but would also reduce the quantity of aniline requiring treatment, which is toxic and difficult to degrade. However, the theoretical yield of 1-D nano-PANI is only 0.80. In order to get 1.0 kg 1-D nano-PANI, in EB form, 1381.2 L volume of reaction solution and 24 h of reaction time would be needed. Because of the low aniline concentration in this theoretical case, the problems associated with excessively high volume of equipment and waste water cannot be overcome. The high cost and large volume of waste water of dilute methods makes it difficult to employ them in commercial production of 1-D nano-PANI at large scale.

The volume of reaction solution was calculated according to Equation (1)
*V*_R_ = *M*_A_/90.5/*Y*_P_/*C*_A0_(1)
where *V*_R_ is the volume of reaction solution in L, *M*_A_ is the mass of PANI in kg, *Y*_P_ is the yield of PANI, *C*_A0_ is the initial concentration of aniline in the reaction solution in mol/L.

The mass of impurities were calculated based on Equation (2)
*M*_I_ = *M*_T_/*D*_T_(*C*_AT_ × 93 + *C*_ST_ × 98 + *C*_AST_ × 132)(2)
where *M*_I_ is the mass of impurities in kg, *M*_T_ is the mass of solution after polymerization in the wet product, *C*_AT_ is the concentration of aniline in the solution after polymerization in mol/L, *C*_ST_ is the concentration of sulfuric acid in the solution after polymerization in mol/L, *C*_AST_ is the concentration of ammonium sulfate in the solution after polymerization in mol/L, *D*_T_ is density of solution after polymerization in kg/L.

The volume of washing water was calculated according to Equation (3)
*V*_w_ = [(*M*_I_/*M*_A_/0.001)^1/3^ − 1] × *M*_T_/*D*_T_(3)
where *V*_w_ is the volume of washing water in L, *M*_I_ is the mass of impurities in kg, *M*_A_ is the mass of PANI in kg, *M*_T_ is the mass of solution after polymerization in the wet product, *D*_T_ is density of solution after polymerization in kg/L.

#### 3.3.3. Rapid Mixing Methods

With rapid mixing methods, because of the low ratio of oxidant to aniline, rapid mixing process can prohibit “secondary growth”, so the processes favor the growth of the 1-D nano-PANI [126,127]. The preparation process of 1-D nano-PANI though rapid mixing methods was presented in Scheme 7. The lower the concentration of aniline, the lower polymerization time [140], the lower molar ratio of oxidant to aniline, and the more sufficient the rapid mixing contribute to the preferential production of 1-D nano-PANI [141]. When the molar ratio of APS-to-aniline was maintained at 1/2, by increasing the concentration of aniline in solution from 8 mM to 128 mM, higher quantities of particulates appeared and the 1-D nano-PANI had more branches and a larger diameter [142]. When the concentration of aniline was 10 mM, with the molar ratio of oxidant to aniline increasing from 0.125 to 1.0, the 1-D nano-PANI was created with a larger diameter. In addition, the mixing process appears to affect the morphology of PANI product during rapid mixing polymerization [143]. As the mixing rate was increased, 1-D nano-PANI with smaller diameter was produced. A rotating packed bed (RPB)—which is a high-gravity device—produced a uniform supersaturated solution that is required for the homogeneous nucleation. This is because the micromixing characteristic time in RPB is much shorter than that of the nucleation induction period in aqueous solutions [46,144]. The PANI nanofibers produced by high-gravity chemical oxidative polymerization (HGCOP) has a more uniform morphology and smaller diameter than those produced in a stirred tank reactor (STR).

The advantages of rapid mixing methods are that no templates are required and the composition of the waste water is relatively simple. However, rapid mixing polymerization has problems in scale-up. The first is that, because the yield of PANI is low, there is a need to degrade a large quantity of aniline. The typical molar ratio of APS to aniline is 1/4 in the rapid mixing methods, which was first reported by Kander et al. [127]. Under these conditions, the theoretical yield of PANI is 0.20. More than 80% of the aniline does not participate in the reaction. To produce 1.0 kg of PANI in the EB form, 4.11 kg of aniline must be degraded or recycled. Although some studies have been done on the increase of 1-D nano-PANI yields, the low yield of 1-D nano-PANI product using rapid mixing methods remains a problem [43]. The low yield produces significant quantities of raw material waste and waste from washing treatment of the product. Recycling of un-reacted raw materials is necessary for commercial production. The second problem is that a uniform supersaturated solution is needed for homogeneous nucleation to produce 1-D nano-PANI. However, the mixing process has an amplification effect, so maintaining uniform rapid mixing in a large volume system is quite difficult.

The low method cost and relatively simple composition of the waste water make the rapid mixing method promising for commercial production of 1-D nano-PANI. However, the problems associated with recycling of raw materials and amplification effect of stirring need to be solved before commercialization of this approach, which need be seriously considered for production of 1-D nano-PANI.

#### 3.3.4. Seeding Polymerization

Adding small amounts of 1-D nano-PANI to a reaction solution can produce high quality PANI nanofibers [145]. It is expected that conventional polymerized PANI and aniline oligomers could also be used as “seed” in the polymerization of PANI [95,126,146,147]. Addition of a “seed” to a reaction mixture is a useful method to improve the morphology of 1-D nano-PANI by chemical polymerization without templates.

### 3.4. Electrochemical Approaches

The electrochemical approaches for synthesis of 1-D nano-PANI include template methods and non-template methods. The templates include porphyrin aggregate [148], block copolymers [149,150], AAO [66,151], etc. 1-D nano-PANI can also be produced directly without a template by galvanostatic electrolysis [15,152,153,154], potentiostatic electrolysis [155,156,157,158], galvanodynamic electrolysis [152,154,159,160], as well as cyclic voltammetry [161,162], pulse-potential method [163,164], potentiodynamic electrolysis [165], etc.

In electrochemical methods, the morphology of the PANI is related to the concentration of aniline, electrolysis potential, potential applied time and manners, temperature, and the medium in non-template methods. With regard to the reaction medium, 1-D nano-PANI can be produced in aqueous and non-aqueous media [166]. However, considering environmental protection and cost, non-aqueous media should not be considered for commercial production of 1-D nano-PANI. It has been found that an increase in potential causes an increase in the radius of the PANI and roughens its surface, because of “secondary growth” [155]. With the exception of the induction period, increasing the period of electrolysis has been found to increase the radius of PANI and roughen the surface of the PANI [15,155]. Increasing the concentration of aniline from 0.03 M to 0.6 M resulted in a satisfactory 1-D nano-PANI [152,155,160,166,167]. A relatively low electrolytic potential, low temperature, as well as a relatively short electrolysis period favor 1-D nano-PANI preparation. In particular, oriented and uniform PANI nanorods arrays were synthesized during a slow growth process. The oriented and uniform PANI nanorods not only have potential applications in field emission devices, but also in supercapacitors and sensors where the oriented and subulate morphology allows for good mass transfer of doping ions [152,154]. However, excessively slow reaction rates limit the application of the electrochemical approach in the production of oriented PANI nanorods.

The properties of PANI are closely related to the electrolytic potential, temperature, and reaction time. The increase in the electrolytic potential and temperature cause that the aniline monomer reacts at the *ortho*- and *meta*- positions and the imines hydrolysis reaction increases. PANI can be degraded by the imines hydrolysis reaction. Therefore, a suitable electrolytic potential, relatively short polymerization time, and low temperature favor synthesis of PANI with a high electronic conductivity and hydrophilicity.

The electrochemical approaches for production of 1-D nano-PANI can be divided into two categories, including potential-controlled electrolysis and current-controlled electrolysis. Because the polymerization of aniline is a self-catalytic process, the electrolytic potential may be too high in the initial stage and too low at the end of polymerization when current controlled electrolysis is employed. Based on the product morphology and electronic conductivity, potentiostatic electrolysis should be one of the suitable methods of the electrochemical approaches for commercial production of 1-D nano-PANI. In a large-scale process, the three-electrode system would be replaced with a two-electrode system to reduce the method costs. The synthesis process of 1-D nano-PANI through electrolysis can be seen in Scheme 8.

There are three advantages of the electrochemical approach. The first is that the waste water of the electrochemical approach has the simplest composition of all the methods used to producing 1-D nano-PANI. The waste water can be relatively easy to be treated and recycled. The second advantage is that the electrolysis process can be easily controlled by changing voltage between two electrodes. The third advantage is that no template and oxidant is used in the electrochemical approach, and the electrolyte solution can be used repeatedly, so high-atom-efficiency could be obtained.

However, there are some problems associated with commercial production of 1-D nano-PANI using the electrochemical approach. For instance, there have been few reports about recycling of raw materials in this process. The yield from single electrolysis bath is low for short polymerization times, but to some extent, electrolyte solution can be used repeatedly. The composition of electrolyte solution after electrolysis has been found to be primarily aniline, HQ, BQ, aniline oligomers, CAB and the doping acids. The HQ, BQ, and CAB will influence the purity, morphology, and properties of PANI. As the electropolymerization time is increased, the concentration of HQ, BQ, and CAB in the electrolyte solution will increase. In the initial electrolysis, the PANI is in uniform nanofiber morphology, as can be seen in Figure 2 [32]. As the electrolysis times are increased, the morphology of the PANI becomes inhomogeneous, as submicron rods and aggregate particles increase in the PANI, as can be seen in Figure 3 [32]. The electrolyte solutions need to be replaced with a fresh electrolyte solution after a certain electrolysis times. The PANI yield never approached to 1.0, so the aniline in the waste electrolyte solution needs to be treated. Recycling of the raw materials from the electrolyte solution is needed for commercial production. A possible way to recycle the used electrolyte solution is to selectively remove the by-products through an extraction process. As a result, the electrochemical properties and morphology of the PANI obtained from recycle electrolyte solution is same to them of the PANI obtained from fresh electrolyte solution [32]. Another issue is the amplification effect on the morphology and properties of PANI. For large scale production, the working electrode area must be increased and the electrolytic cells should be connected in series. The distribution of current, voltage, and temperature strongly affect the morphology, electronic conductivity, and hydrophilicity of the PANI products. Electrochemical polarization occurs in the cell where the distribution of current on the working electrode becomes increasingly non-uniform so that the current concentrates at the edge of the parallel electrodes as the working electrode area increases [168]. The PANI produced at different positions of electrode occurs at different current densities. Using working electrodes with area increasing from 0.08 m × 0.085 m to 0.10 m × 0.50 m, PANI products had a nanofiber shape with relatively uniform size, as seen in Figure 4a–c [169]. However, as the working electrode area was increased to 0.10 m × 1.00 m, the morphology of PANI became inhomogeneous, with submicron rods and aggregate particles appearing on the PANI, as seen in Figure 4d. This indicates that the working electrode area cannot be increased indefinitely. In addition, a high and concentrated current results in production of side products generating through reactions at *ortho*- and *meta*- position as well as the imine hydrolysis reaction. The side reactions will affect the morphology and properties of the PANI. Too large a working electrode area will result in inhomogeneity of the PANI products, a waste of electrode area, and a decrease of the electronic conductivity and purity of the product. Secondly, as the number of electrolytic cells in series increases, the voltage and temperature of each cell is difficult to maintain at a uniform level. Nonuniform cell voltages result in unsuitable electropolymerzation potentials. The electropolymerization of aniline is exothermic, so heat accumulates in the center of the electrolytic cells. Unsuitable eletropolymerzation potentials and high temperatures result in PANI with uniform morphology and low electronic conductivity. Therefore, as the number of cells in series increases, the quality of the product decreases. Due to this amplification effect, a more reasonable design of the electrolytic cells is necessary for commercial production of 1-D nano-PANI, such as cylindrical or spherical electrode and electrolytic cell. Also, more suitable electrode materials are needed for large scale production of 1-D nano-PANI. Currently, electrode materials include graphite, stainless steel, platinum, gold, etc. Stainless steel is a common electrode material for the producing of 1-D nano-PANI, due to the good electrochemical inertia, cheap price, and good electronic conductivity. However, it is reported that PANI can passivate stainless steel which induces a high resistance of polymerization. Therefore, new electrode materials for commercial production of 1-D nano-PANI should be selected. Platinum and gold electrodes are too expensive to be practical. Graphite electrodes are relatively promising due to the high electronic conductivity and good electrochemical inertia. However, graphite electrodes are soft, and mechanical stripping of the product will lead to a loss of electrode materials. Therefore, modification of existing electrodes is needed. Potentiostatic electropolymerization is a promising method for the producing of 1-D nano-PANI, because of its high atom-efficiency, easy control of the reaction, and easy treatment of waste. The problems for this method include, recycle treatments, amplification effects, and electrode selections, which need to be addressed for commercial production of 1-D nano-PANI.

### 3.5. Other Methods

Other methods for producing 1-D nano-PANI include sonochemical synthesis, solid-state polymerization, UV light-assisted polymerization, radiolytic synthesis, microwave-assisted polymerization, plasma-induced polymerization, and electrospinning. The sonochemical synthesis, UV light-assisted polymerization, radiolytic synthesis, microwave-assisted polymerization, and plasma-induced polymerization require ultrasonic, UV, radiolytic, microwave, or plasma devices. These devices expend considerable energy and are costly. In addition, it is difficult to uniformly distribute the ultrasonic wave, UV light, ray, microwaves, or plasma uniformly in a larger-scale reactor. This is the main reason that limits their commercial production of 1-D nano-PANI using these techniques. Since no additional surfactants or organic solvents are needed in solid-state polymerization, the method is shown to be simple and easy [170,171]. However, the product yield was low in solid-state polymerization and grinding will produce a lot of heat which would increase by-reactions. Electrospinning is a simple way to obtain polymer nanofibers. However, it is almost impossible to produce pure 1-D nano-PANI via electrospinning.

## 4. Conclusions

1-D nano-PANI has very promising commercial applications such as sensors, electrochromic devices, supercapacitors, anticorrosive coatings, etc. Therefore, commercial production of 1-D nano-PANI at large scale is urgently needed. The main synthesis methods of 1-D nano-PANI are the hard template methods, soft template methods, and chemical template-free methods including interfacial polymerization, rapid mixing polymerization and dilute polymerization, and electrochemical approaches. Although aniline is cheap and the polymerization process is simple, its monomer linkage position may change and polyaniline may depredate under certain conditions. Meanwhile, aniline is toxic and difficult to degrade. Therefore, the treatment of discharged waste waters from all preparation methods is tedious and difficult. We do think that the treatment and recycling of discharged liquids should be the first problem for commercial production of 1-D nano-PANI at large scale. The comparison among different processes of the aspects of composition complexity of waste liquid, volume of waste liquid, waste quantity, and recycling complexity of waste liquids—as well as method costs—can be seen in Table 1. Due to the high treatment complexity of waste liquid which has extremely complex composition and high method cost, hard template methods are not the most suitable methods for commercial production of 1-D nano-PANI. Due to the high treatment complexity of waste liquid, soft template methods and interfacial polymerization are also not the most suitable methods. Due to the large volume of the waste water and the required equipment, dilute methods are not the most suitable methods for commercial production. Because of the easy treatment and recycling of waste water with a relatively simple composition, rapid mixing and electropolymerization methods might be the most suitable approaches for commercial production of 1-D nano-PANI. However, there are problems in these processes that need to be solved before commercial production can be realized. For rapid mixing methods, the recycling process of the raw materials (low yield) and the scale-up effects of mixing need to be solved. For electropolymerization, the recycling of raw materials (low yield), scale-up effects of electrode reaction and the selection of suitable electrode materials need to be solved.

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
