# Peer review of "Some Important Issues of the Commercial Production of 1-D Nano-PANI"

_polymers, 2019, doi:10.3390/polym11040681_

Round 1
Reviewer 1 Report
The review article is about the synthesis of 1-D nano-PANI in the perspective of mass production and waste production.Each methods are presented with advantages and non avantages in terms of costs and waste production and treatment.
Some parts of the article contain large enumeration of compounds or methods ( line 131 to 146) or( lines 194 to 213) for exemples which is hard to read.
My question is how this article is usefull for scientists in the field? it also misses the relation between synthsis methods and physical or chemical properties of the materials.
I'm not english native but it seems to me that many mystakes should be corrected.
In my opinion , this article should not be published in the journal "Polymers"
Author Response
Dear Reviewer:
Thank you for your letter and for the reviewer’s comments concerning our manuscript entitled “Some Important Issues of the Commercial Production of 1-D nano-PANI” (polymers-440557). Those comments are all valuable and very helpful for revising and improving our paper, as well as the important guiding significance to our researches. We have studied comments carefully and have made correction which we hope meet with approval. Revised portion are highlighted in the paper. The main corrections in the paper and the responds to the reviewer’s comments are as flowing:
Q: Some parts of the article contain large enumeration of compounds or methods ( line 131 to 146) or( lines 194 to 213) for exemples which is hard to read.
A: The parts contains large enumeration of compounds or methods are rewritten. Mistakes in English have been reexamined and corrected。
Q: My question is how this article is usefull for scientists in the field
A: We do think that the goal of scientific research should be commercial application of the mentioned substance or technology at large scale. In the last 40-year, large numbers of papers have been published on preparation, characterization and application of polyaniline one-dimensional nanomaterials(1D Nano-PANI) and their applications which are close related to their electrochemical properties, reversible redox behavior, machinability, conductivity and so on. Also, its applications in sensors and actuators, electromagnetic shielding, conductive paints, energy storage components, protective coatings, photoelectric devices and separating membranes have been reported or reviewed. But, almost have no report on preparation of1D Nano-PANI commercially at large scale. If one want to use 1D Nano-PANI commercially at large scale, some important issues must be considered. We also do think that the solutions related to these issues not only an engineering problem, but a scientific problem. For example, how to prohibit or inhibit the side reaction in 1D Nano-PANI preparation process is an important issues for scientists. Here, we overview the methods which are related 1D Nano-PANI preparation based on waste liquid treatment and recycling of production 1D Nano-PANI. The aim of this article is to help scientists and engineers comprehensive understanding of polyaniline, develop convenient process to produce 1D Nano-PANI commercially in large scale.
Q: it also misses the relation between synthsis methods and physical or chemical properties of the materials.
A: Just as above mentioned, there are large numbers of papers have been published on preparation, characterization and application of polyaniline one-dimensional nanomaterials(1D Nano-PANI) and their applications which are close related to their electrochemical properties, reversible redox behavior, machinability, conductivity and so on. The physical or chemical properties of 1D Nano-PANI are not only related with methods,but close related to preparation conditions of each methods including oxidant potential, concentration, temperature, templates, doping acid and so on. It is too complicated to summarize. This article focuses on the analysis of important issues which limit the commercial application of 1D Nano-PANI. If the reviewer do think this is very important, we can do our best to try add this content in our paper.
Special thanks to you for your good comments.
We tried our best to improve the manuscript and made some changes in the manuscript.
We appreciate for Reviewers’ warm work earnestly, and hope that the correction will meet with approval.
Once again, thank you very much for your comments and suggestions
Yours sincerely,
Ying Wu

Reviewer 2 Report
This manuscript provides a comprehensive review on the synthesis methods of 1-D nanomaterials, including nanofibers, nanowires, nanotubes, nanobelts and nanorods, of polyaniline (PANI). The authors first describe the complexity of the polymerization of PANI, in which the problems associated with the treatment of the waste chemicals from the polymerization are described. The authors considered the difficulty in treatment and recycle of the waste liquid is the major obstacles to the commercial production of 1-D nano-PANI. In the section part, the authors reviewed comprehensively the synthesis methods for the production of 1-D PANI, including template methods and non-template methods. Finally, the authors considered the complexity of the composition of the waste liquid and the cost or equipment, the authors suggested that the rapid mixing methods and electropolymerization methods might be the most suitable approched for commercial production 1-D nano-PANI, although a number of problems still exist.
Overall, I think this is a nice review article with a well-defined topic for an important polymer with great potential in applications. The review is comprehensive, covering essentially all reported synthesis of 1-D PANI. The references are also well organized and presented in the text. Although the English writing requires some improvement, the manuscript is basically well written. Therefore, I would recommend the manuscript for publication in Polymers. I only have the following minor comments which require the consideration for improving the manuscript.
1. I would like to suggest the title to be changed to “Some Important Issues of the Commercial Production of 1-D Polyaniline Nanomaterials”. The present title is somewhat awkward.
2. A keyword of the manuscript is “commercial production”. The authors should hence describe/review the current status of the commercial production of PANI.
3. Table 1 presents the comparison among various methods on the waste liquids composition complexity, waste quantity, recycle complexity, etc. The comparison is purely quanlitative, which makes it rather unconvincing. I am thinking if it would be possible to do it more quantitatively, if relevant information can be found in literature.
Author Response
Dear Reviewer:
Thank you for your letter and for the reviewer’s comments concerning our manuscript entitled “Some Important Issues of the Commercial Production of 1-D nano-PANI” (polymers-440557). Those comments are all valuable and very helpful for revising and improving our paper, as well as the important guiding significance to our researches. We have studied comments carefully and have made correction which we hope meet with approval. Revised portion are highlighted in the paper. The main corrections in the paper and the responds to the reviewer’s comments are as flowing:
Q:I would like to suggest the title to be changed to “Some Important Issues of the Commercial Production of 1-D Polyaniline Nanomaterials”. The present title is somewhat awkward.
A: Title have changed.
Q: A keyword of the manuscript is “commercial production”. The authors should hence describe/review the current status of the commercial production of PANI.
A: Just as Reviewer suggested, the current status of the commercial production of PANI has been described.
To our best knowledge, at present, bulk polyanilines are commercially produced by chemical oxidation process, and are mainly used in anticorrosive coatings, antistatic coatings, etc. The companies are world spread, such as Ormecon (Germany), Ancatt (USA), Zhongke Benan (Hunan, PRC), etal and many other companies are actively developing the industrial application of polyaniline. The corresponding contents were added in our revised manuscript.
Q: Table 1 presents the comparison among various methods on the waste liquids composition complexity, waste quantity, recycle complexity, etc. The comparison is purely quanlitative, which makes it rather unconvincing. I am thinking if it would be possible to do it more quantitatively, if relevant information can be found in literature.
A: The method cost is anal sized based on the complexity of the method, reaction volume (equipment cost and operation cost) and raw materials used in the preparation one-dimensional polyaniline nanomaterials (1-D nano-PANI). Templates and the huge reaction volume will increase the cost.
The difficult in reusing of waste water depends on the composition complexity of discharged waste water. Even though may be called as the same method, their preparation condition may be has wide a range of change, furthermore, the yield data are very lack in literatures. We will do our best to calculate the amount of waste liquid of typical methods. But, it is difficult to perform comprehensive quantification among them.
For example, hard template includes porous carbonate membrane (prepared by heavy ions accelerated by synchrotron radiation and the pore number may be range from 107 to 109) whose effective thickness is about 4 μm, anodic aluminum oxide (prepared by electrochemical method and the pore number may be as high as 1011) whose effective thickness may be depth as 10μm, et al, under these condition, 1-D nano-PANI can only grow in pore; the pore volume is only a small part compare to the volume of the whole materials, so the yield of the 1D-PANI-Nano is very low. When other one-dimensional materials such as carbon nanotube, MnO2 nanowire or V2O5 nanowire were used as hard template, their cost may be changed by their raw materials and the process how to prepare them.
When soft-template-methods were used to prepare 1D-PANI-Nano, the yield might be as high as 80%[1], while the discharged waste water may contain aniline (at least about 20% of original aniline added as reactant) , structural direct agent, salts, acids, by products. So the recycle complexity of waste water recycle is high.
Interfacial polymerization method used for preparing 1-D nano-PANI was first reported by Kaner et al.[2, 3]. One typical condition was 1 L of aqueous solution with 0.5−1 M acid and 200 mL of organic solvent with 1.6 M aniline were used [2, 3]. The concentration of ammonium peroxydisulfate (APS) are 0.08M. The aniline to ammonium peroxydisulfate molar ratio was kept at 4 to 1. the yield was in the range of 6-10% [2, 3]. If we want to obtain 1 Kg PANI (EB), at least 414 L waste liquid (Reaction solution volume) and 69 L organic solvents will discharged. In the waste liquid will contain aniline (at least about 90% of original aniline added as reactant), salt, doping acid, aniline, by-products and organic solvent containing some by-products. So, the discharged liquid is very complex and it is very difficult to use the charged liquid repeatly.
Dilute polymerization also may be used to prepare 1-D nano-PANI, which was first reported by Epstein[4]. The typical process as follow: aniline is 8 mM and the doping acid is 1.0 M, the reaction time is 24 h, the APS to aniline molar ratio is at 1/2. So, the theoretically yield is less 40%. The discharged waste liquid for 1.0 kg 1-D nano-PANI is at least 3453.2 L (Reaction solution volume) containing aniline (at least about 60% of original aniline added as reactant), acid, salt and by-products.
Rapid mixing methods may be thought as a suitable method to prepare 1-D nano-PANI. The typical preparation conditions as follow: APS to aniline is 1/4, aniline is 0.16 M, and the doping acid is 1.0 M [5]. The theoretically yield is 20%. If one want to obtain 1.0 kg 1-D nano-PANI, at least 345.3 L liquid containing aniline (at least about 80% of original aniline added as reactant), doping acid, aniline, salt, by-products will be discharged. Although the discharged volume was decreased, its total amount waste such as aniline was increased.
Electrochemical approaches are related to various potentials applied methods. Its disadvantage is its low yield (usually lower than 10% for a single electrolysis process). Its advantage is the discharged liquid is very simple, only contain aniline, acid and small amount of by-products. Here, we select our electropolymerization result to calculated[6].
The typical preparation conditions are as follow: the cells can contain 100 ml electrolyte, the electrode area is 0.01 m2, temperature 293 K, reaction time 20 min, cell voltage 1.58 V. The yield is about 8%. If we want to produce 1.0 kg 1-D nano-PANI, 552.5 L used electrolyte containing aniline (at least about 92% of original aniline added as reactant), acid, and by-products. Now, we have developed a method to purification the used electrolyte and to reused it[7].
[1]Wu T. M., Lin Y. W., Synthesis and characterization of hollow polyaniline microtubes and microbelts with nanostructured walls in sodium dodecyl sulfate micellar solutions, Polymer Engineering and Science, 2008, 48(4): 823-828.
[2]Huang J. X., Virji S., Weiller B. H., et al., Polyaniline nanofibers: Facile synthesis and chemical sensors, Journal of the American Chemical Society, 2003, 125(2): 314-315.
[3]Huang J. X., Kaner R. B., A general chemical route to polyaniline nanofibers, Journal of the American Chemical Society, 2004, 126(3): 851-855.
[4]Chiou N. R., Epstein A. J., Polyaniline nanofibers prepared by dilute polymerization, Advanced Materials, 2005, 17(13): 1679-+.
[5]Huang J. X., Kaner R. B., Nanofiber formation in the chemical polymerization of aniline: A mechanistic study, Angewandte Chemie-International Edition, 2004, 43(43): 5817-5821.
[6]Zhang Haibin, Preparation and characterization of polyaniline nanomaterials; Tianjin University
[7]Wu Y., Wang J., Ou B., et al., Electrochemical Preparation of Polyaniline Nanowires with the Used Electrolyte Solution Treated with the Extraction Process and Their Electrochemical Performance, Nanomaterials, 2018, 8(2).
Special thanks to you for your good comments.
We tried our best to improve the manuscript and made some changes in the manuscript.
We appreciate for Reviewers’ warm work earnestly, and hope that the correction will meet with approval.
Once again, thank you very much for your comments and suggestions
Yours sincerely,
Ying Wu

Round 2
Reviewer 1 Report
I consider that the authors have significantly improved the quality of their article and as a result the article is now ready for publication.
I suggest only a small change : in the lines 212 -214 the DBSA as surfactant is repeated twice.
Author Response
Dear Reviewer:
Thank you for your letter and for the reviewer’s comments concerning our manuscript entitled “Some Important Issues of the Commercial Production of 1-D nano-PANI” (polymers-440557). Those comments are all valuable and very helpful for revising and improving our paper, as well as the important guiding significance to our researches. We have studied comments carefully and have made correction which we hope meet with approval. The main corrections in the paper and the responds to the reviewer’s comments are as flowing:
The mistake in 212 -214 have been modified. And the mistakes in grammar have been reexamined and corrected.
Thanks to you for your good comments.
We tried our best to improve the manuscript and made some changes in the manuscript.
We appreciate for Reviewers’ warm work earnestly, and hope that the correction will meet with approval.
Once again, thank you very much for your comments and suggestions
Yours sincerely,
Ying Wu